# ViNT: A Foundation Model for Visual Navigation

**Dhruv Shah[†], Ajay Sridhar[†], Nitish Dashora[†],**
**Kyle Stachowicz, Kevin Black, Noriaki Hirose, Sergey Levine**
UC Berkeley

**Abstract:** General-purpose pre-trained models ("foundation models") have enabled practitioners to produce generalizable solutions for individual machine learning problems with datasets that are significantly smaller than those required for learning from scratch. Such models are typically trained on large and diverse datasets with weak supervision, consuming much more training data than is available for any individual downstream application. In this paper, we describe the Visual Navigation Transformer (ViNT), a *foundation model* that aims to bring the success of general-purpose pre-trained models to vision-based robotic navigation. ViNT is trained with a general goal-reaching objective that can be used with any navigation dataset, and employs a flexible Transformer-based architecture to learn navigational affordances and enable efficient adaptation to a variety of downstream navigational tasks. ViNT is trained on a number of existing navigation datasets, comprising hundreds of hours of robotic navigation from a variety of different robotic platforms, and exhibits *positive transfer*, outperforming specialist models trained on narrower datasets. ViNT can be augmented with diffusion-based goal proposals to explore novel environments, and can solve kilometer-scale navigation problems when equipped with long-range heuristics. ViNT can also be adapted to novel task specifications with a technique inspired by prompt-tuning, where the goal encoder is replaced by an encoding of another task modality (e.g., GPS waypoints or turn-by-turn directions) embedded into the same space of goal tokens. This flexibility and ability to accommodate a variety of downstream problem domains establish ViNT as an effective foundation model for mobile robotics.

**Keywords:** visual navigation, multi-task learning, planning, generalization

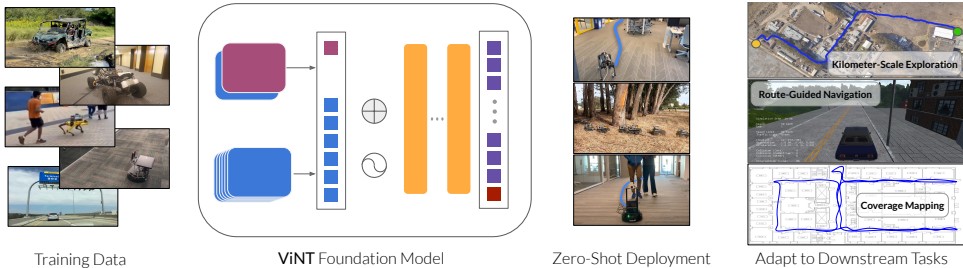

Training Data          ViNT Foundation Model          Zero-Shot Deployment          Adapt to Downstream Tasks

**Figure 1: Overview of the ViNT foundation model.** ViNT generalizes zero-shot across environments and robot embodiments, and can be directly applied to tasks including exploration and navigation around humans. ViNT can also be fine-tuned with a small amount of data to expand its capabilities to new tasks.

## 1 Introduction

Recently, machine learning methods have achieved broad success in natural language processing [1], visual perception [2–4], and other domains [5, 6] by leveraging Internet-scale data to train general-purpose "foundation" models that can be adapted to new tasks by zero-shot transfer, prompt-tuning, or fine-tuning on target data [7–10]. Although this paradigm has been successful in many domains,

---

[†] Lead Authors. Unabridged paper, videos, and code: general-navigation-models.github.io.

7th Conference on Robot Learning (CoRL 2023), Atlanta, USA.

it is difficult to apply in robotics due to the sheer diversity of environments, platforms, and applications. In this paper we ask the question: *what is required of a foundation model for mobile robotics?*

In this paper, we define a *robot foundation model* as a pre-trained model that can be (i) *deployed zero-shot* in novel, useful settings (e.g., different sensors, robot embodiments, environments etc.), and (ii) *adapted* to a downstream task of choice (e.g., different objectives, goal specification types, behaviors etc.). We specifically consider the problem of visual navigation, where the robot must navigate its environment solely using egocentric visual observations. A general pre-trained robot navigation model should enable a wide range of navigation applications, readily allow fine-tuning to downstream tasks, and generalize to a broad range of environments and robotic platforms. Such a model should provide a broadly capable navigation policy on top of which applications to specific domains can be constructed, giving a base level of generalization and capabilities to new robotic platforms in zero shot that can be further improved after fine-tuning with a small amount of data.

To this end, we propose the **Vi**sual **N**avigation **T**ransformer, or ViNT: a cross-embodiment foundation model for visual navigation with strong zero-shot generalization. We train ViNT to reach goals specified by camera images, providing a very general pre-training objective that can be applied to almost any mobile robot dataset. We propose a novel exploration algorithm for the visual navigation paradigm using a diffusion model to propose short-horizon goals, and demonstrate that it enables ViNT to navigate in novel environments. ViNT can control new robots in zero-shot, explore previously unseen environments, perform indoor mapping, and navigate kilometer-scale outdoor environments without interventions. Furthermore, we show that ViNT can be fine-tuned on a small amount of data to achieve high performance with new task specification modalities – such as GPS waypoints or high-level routing commands – allowing ViNT to serve as a foundation for a wide variety of navigation applications. Lastly, we qualitatively analyze some emergent behaviors exhibited by ViNT, such as implicit preferences and navigation around dynamic pedestrians.

We hope that ViNT represents a step towards such general-purpose *robot foundation models* that can be deployed on a wide range of robots, and on a wide range of tasks, and serve as a foundation for diverse mobile robotic applications. Model weights for ViNT as well as training and deployment code will be released on our project page: general-navigation-models.github.io.

## 2 Related Work

Learning from large, diverse robotic datasets has been studied for various robotic applications where data sharing across *similar* robots provides a larger training set for more generalizable models [11–13]. However, for applications in mobile robotics, with varying dynamics and camera configurations (e.g., focal length, field of view, and extrinsics), current approaches tend to rely on learning either from small real-world datasets, which are only representative of a single robotic platform, or from simulation, with paired robot and environment models to transfer learned policies [14–16]. Instead, our paper follows the paradigm of learning navigation behavior from data collected across multiple different real-world robotic systems [17–19], while focusing on training a foundation model that can be adapted for a variety of downstream tasks in zero shot or with small amounts of data.

Our goal is to train an effective visual navigation policy that can solve a range of downstream tasks, such as navigating to GPS goals [20], goal images [21], and skill-conditioned driving [22].Following a large body of research in visual navigation, we use a combination of topological graphs for maintaining a spatial representation of the environment and learned policies for low-level control [23–28], and use learned heuristics to guide the robot in novel environments [15, 29]. But unlike these works, our goal is to train a single generalist model rather than specialist solutions to each of these problems, showing how a single high-capacity model can be adapted for diverse tasks.

The closest related works to ViNT are RT-1, I2O, and GNM [15, 19, 30], which study broad generalization across environments and embodiments for robots deployed in real-world settings. While RT-1 demonstrates impressive performance in following diverse instructions, our focus is on adapting a single model across *many* robots to solve *different tasks*, by fine-tuning with small amounts of data. I2O and related efforts [15, 16] show impressive transfer from simulation to real-world envi-

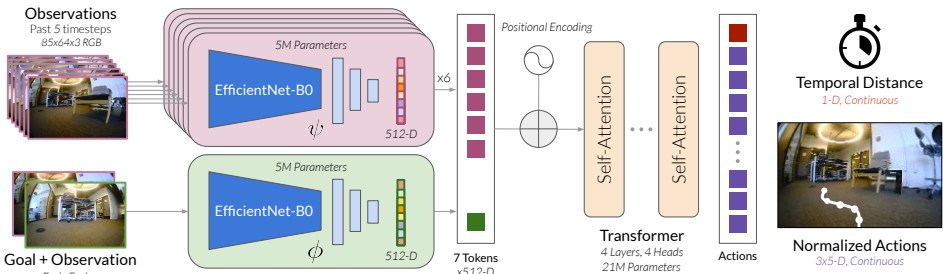

**Figure 2: ViNT Model Architecture.** ViNT uses two EfficientNet encoders $\psi, \phi$ to generate input tokens to a Transformer decoder. The resulting sequence is concatenated and passed through a fully-connected network to predict (temporal) distance to the goal as well as a sequence of $H = 5$ future actions.

ronments, but we emphasize that our aim is orthogonal to the specific choice of algorithm: we focus on learning a capable navigation policy that can be efficiently adapted to solve different downstream tasks. GNM [19] demonstrates policy learning from heterogeneous RGB datasets, but focuses on the singular task of reaching image goals in the zero-shot setting. Instead, ViNT trains a single generalist policy with an emphasis on adaptation to new embodiments and tasks in downstream applications, though it can also be used zero-shot to great effect (Sec. 6.1).

## 3 The ViNT Model

Our model is trained for image-goal navigation, providing general navigational capabilities that can then either be utilized directly, or serve as a pre-trained foundation for downstream fine-tuning with other task specifications. In the image-goal navigation task, the robot is tasked with navigating to a subgoal specified by an image observation $s$ (i.e., the robot's observation at the goal). Unlike alternative mechanisms for goal specification such as PointGoal [31], GPS navigation, or semantic objectives [32], a model can be trained for image-goal navigation with minimal assumptions, utilizing any data that contains videos and actions, without requirements on ground-truth localization, semantic labels, or other metadata. This makes it practical to train on a large and diverse dataset sourced from many different robots, facilitating broad generalization.

ViNT takes as input current and past visual observations $o_{t-P:t}$ and a subgoal image $o_s$, and predicts (i) the number of time steps needed to reach the subgoal (the *dynamical* distance), and (ii) a sequence with length $H$ of future actions leading towards the subgoal. Our 31M-parameter model, ViNT, is built on the Transformer architecture [33] and is optimized for: (i) fast and efficient inference on resource-constrained robots, and (ii) the ability to prompt and fine-tune for downstream tasks. We initialize all networks from scratch and train them end-to-end with a maximum likelihood objective. The model architecture is summarized in Figure 2, and described in detail in Appendix A.

**Tokenization:** The ViNT architecture (Fig. 2) first tokenizes its inputs into an embedding of size $d_{\text{model}} = 512$. ViNT independently tokenizes current and $P = 5$ past visual observations by encoding them with an EfficientNet-B0 [34] model, which takes $85 \times 64 \times 3$ images as input and outputs a flattened feature vector $\psi(o_i)$ from the final convolutional layer [30].

**Goal fusion:** We found that naïvely extracting features from the goal image $\phi(o_s)$ using an EfficientNet encoder $\phi$ led to poor performance, often ignoring the goal entirely (see Appendix A). We hypothesize that effective features for image-based goal-reaching tasks are often *relative*, encoding the *difference* between the current observation and the goal rather than an absolute representation of the goal itself. Hence, we use a separate *goal fusion* encoder $\phi(o_t, o_s)$ to jointly encode the current and goal observations. We stack the two images along their channel dimensions, pass them through a second EfficientNet-B0 encoder, and flatten to obtain the goal token.

**Transformer:** The $P + 2$ observation and goal tokens are combined with a positional encoding and fed into a Transformer backbone $f$. We use a decoder-only Transformer with $n_L = 4$ multi-headed attention blocks, each with $n_H = 4$ heads and $d_{\text{FF}} = 2048$ hidden units.

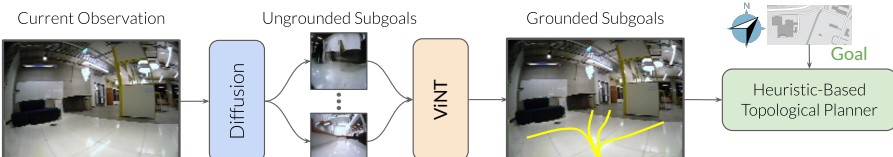

**Figure 3: Long-horizon navigation in unseen environments with ViNT.** We use physical search with a topological graph-based planner to explore the environment. An image-to-image diffusion model proposes diverse exploration targets which are spatially grounded using ViNT (yellow), and scored using a goal-directed heuristic $h$. Subgoals are added to the topological graph $\mathcal{M}$ and executed using the ViNT policy.

**Training data:** We train ViNT using a large-scale dataset of heterogeneous navigation trajectories from a diverse set of environments and robotic platforms with varying dynamics, camera parameters, and behaviors. The training dataset contains over 100 hours of real-world trajectories sourced entirely from existing datasets, spanning 8 distinct robotic platforms with varying speeds and dynamics. For more details about the dataset, see Appendix C.

**Embodiment-agnostic action space:** To effectively train a single model across robots of varying sizes, speeds, and dynamics, we follow Shah et al. [19] and choose an embodiment-agnostic action space for ViNT. To abstract away low-level control, ViNT uses relative waypoints as its action space $\hat{a}$; to account for the large variance in speeds and sizes of the robots, we normalize these waypoints by scaling them according to the robot's top speed. During deployment, a robot-specifc controller is used to un-normalize and *track* these waypoints using low-level control.

## 4 Long-Horizon Navigation with ViNT

While the goal-conditioned policy learned by ViNT captures a general understanding of navigational affordances and obstacles, it has limited applicability on its own. Many practical tasks are either not defined by goal images, or require a much longer horizon than what ViNT directly supports. We apply ViNT to several downstream applications by combining it with an episodic memory formed by a topological graph, which provides short-horizon subgoals for reaching faraway locations. In previously unseen environments, we can further augment this graph-based planner with exploratory subgoal proposals, which can drive ViNT to explore a new environment and discover a path to the goal. We consider multiple such proposal mechanisms and find that maximum performance is attained by an *image diffusion model* that samples diverse future subgoal candidates conditioned on the current observation.

These subgoals are scored with a goal-directed heuristic to identify the best subgoal that makes progress towards the goal using a process akin to *physical A\* search* [29]. Past observations and unexplored frontiers are stored as nodes in a topological graph, with their connectivity determined by the distances predicted by ViNT. During exploration, we build this topological graph on-the-fly as the robot explores the environment. During later deployments it may be used for discovering shortcuts to arbitrary goals in the environment. We first describe the high-level algorithm that plans on top of subgoal candidates, and then discuss the process for obtaining these subgoal candidates.

### 4.1 High-Level Planning and Exploration

Let's assume that we have access to subgoal candidates $o_{s_i} \in \mathcal{S}$ available to ViNT for planning. We incorporate these subgoal candidates into an exploration framework for goal-directed exploration in novel environments, where the user provides a high-level goal $G$, which may be arbitrarily far away. We largely follow prior work [29], but swap out the learned models with ViNT and the diffusion model. We summarize the system here, and provide a more complete discussion in Appendix B.3.

We construct a topological graph $\mathcal{M}$ online to act as episodic memory, with each node as an individual subgoal observation and edges representing paths between two subgoals, added when the path is taken by the robot, or the model predicts a subgoal to be *reachable* from another node. We frame goal-directed exploration as a search problem, where the robot incrementally builds $\mathcal{M}$ while searching for the goal. To guide search towards the goal, the robot uses a goal-directed heuristic

$h(o_t, o_{s_i}, G, \mathcal{M}, C)$ to *score* subgoal candidates according to their likelihood of reaching the goal, given additional context $C$ — for example, a floor plan or satellite imagery [15, 29]. This heuristic may be geometric (e.g., Euclidean distance) or learned (see Appendix B.3).

During deployment in a new environment, the robot uses the diffusion model to generate subgoal candidates $\mathcal{S}$ from $o_t$, spatially grounds them using ViNT, and scores them using the goal-directed heuristic $h(.)$. The robot then selects the best subgoal $o_{s*}$ according to this heuristic using an $A^*$-like planner, adds it to $\mathcal{M}$, and drives towards it using ViNT (Figure 3). During subsequent deployments in the same environment, the robot can use $\mathcal{M}$ to discover shortcuts to arbitrary goals in the environment. Please see Appendix B.3 for more details about the planner and heuristics. In our experiments, we consider two candidate search heuristics: a geometric heuristic based on positions of the robot and the goal, and a learned heuristic based on additional context in the form of a satellite image.

## 4.2 Subgoal Generation with Diffusion

The physical search algorithm presented above relies on the ability to propose subgoal candidates $\mathcal{S}$ that are both diverse and reachable from the current observation of the robot $o_t$. This amounts to sampling from a high-dimensional, multimodal distribution of RGB images.

To do so, we train a conditional generative model $g(o_{s_i}|o_t)$ on the ViNT training data. Specifically, we apply an image-to-image diffusion model [35, 36], a generative model class that is well-suited for producing diverse samples over high-dimensional spaces such as RGB images. We train the model using randomly-sampled future observations from trajectories in the ViNT dataset (Appendix B.2), and sample $K$ subgoal candidates $\mathcal{S} = \{s_1, \ldots, s_K\}$ from the model at inference time.

However, these subgoal generations are not *spatially grounded*: they do not include an actionable relationship to $o_t$. We ground these candidates by using ViNT to compute temporal distances $d(s_i, o_t)$ and action rollouts $a(s_i, o_t)$, yielding a set of grounded subgoals as in Fig. 14. While the samples generated by the diffusion model do not necessarily match any real observation (see Figure 3), they preserve sufficient relative features from $o_t$ to be plausible, and we find that ViNT generalizes well to generated subgoals. We further study the behavior of this diffusion model in Section F.

## 5 ViNT: A Foundation Model For Downstream Tasks

Beyond its core functionality as an image goal-conditioned model, we show that the strong navigational priors learned by ViNT can be adapted to a variety of downstream tasks, beyond image goals, by fine-tuning part or all of the model in novel environments or with new modalities of data.

**Full model fine-tuning:** While ViNT demonstrates strong zero-shot generalization to new environments and robots, we can further improve on-task performance by fine-tuning the entire model with the same objective but using on-task data. This allows ViNT to quickly learn new skills, forming a continually improving model. ViNT can master new environments and embodiments with as little as 1 hour of navigation data, transferring the capabilities of the original model to a new setting without retraining from scratch.

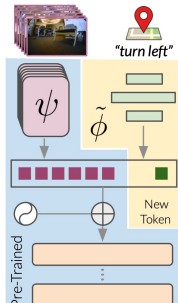

**Adapting to new modalities:** While specifying image goals gives a general pre-training objective, ViNT can easily be *adapted* to other forms of goal-specification by learning a "soft prompt" mapping from the desired goal modality to the ViNT goal token [10]. We build on the Transformer architecture's ability to attend to multimodal inputs projected into a shared token space [37, 38]. Given a subgoal $\sigma$ in a new modality (such as 2D coordinates or routing directions [22]),

**Figure 4:** Adapting ViNT to different goals using a tunable goal token.

we train a small neural network $\tilde{\phi}$ that maps the subgoal to this shared token space as shown in Figure 4, and replace $\phi(o_t, o_s)$. This allows adaptation to new tasks with minimal data, while still leveraging the performance and generalization of ViNT. Appendix B.4 includes additional details.

|                          | **Indoor** | **Outdoor** |
|--------------------------|:----------:|:-----------:|
| Method                   | Success    | Success     |
| End-to-End BC            | 0.72       | 0.44        |
| End-to-End GCG [39]      | —          | 0.61        |
| RECON [40]               | 0.19       | 0.23        |
| ViNT-R (Random Subgoals) | 0.81       | —           |
| ViNT                     | **0.94**   | **1.00**    |

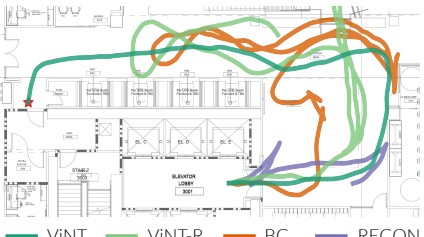

**Table 1:** ViNT paired with our physical search algorithm consistently outperforms baselines for the task of undirected goal-reaching in indoor and outdoor environments (*left*). By effectively planning over diffusion subgoal proposals, ViNT is able to find an efficient path to the goal. Other baselines struggle to explore large indoor environments, shown by trajectories overlaid on an indoor floor plan (*right*).

## 6 Real-world Evaluation

We deployed our ViNT foundation model on five distinct robotic platforms, including a drone, a quadruped, and two other *novel* robots which are not present in the training data. We designed our experiments to answer the following questions:

**Q1.** Can ViNT efficiently explore previously unseen environments and incorporate heuristics?

**Q2.** Does ViNT generalize to *novel* robots, environments, and obstacles?

**Q3.** Can ViNT be fine-tuned to improve performance in out-of-distribution environments?

**Q4.** Can the ViNT policy be adapted to handle new task specification and modalities?

Please see Appendix D for more details on platforms used in the training data and in evaluation.

### 6.1 Navigation Performance

Towards understanding **Q1**, we deploy our full graph-based navigation pipeline (Section 4.1) in a variety of challenging indoor and outdoor environments, previously unseen in the training data. We evaluate the performance of ViNT on two challenging tasks: (i) coverage exploration, where the objective is maximally explore an environment in search of a goal whose location is unknown, and (ii) guided exploration, where the objective is to reach a goal using contextual information such as GPS coordinates or a satellite image (see Figure 11 for task examples). We compare ViNT to a variety of baselines, including end-to-end policies trained with imitation or RL [15, 39], a prior graph-based approach using a VIB for exploration [40], and an ablation of ViNT that samples random images from the training set to use as goals rather than generating subgoals with a diffusion model. See Appendix E.1 for details about the experimental setup.

For the coverage exploration task, the agent is placed in an unknown environment and tasked with exploring the environment maximally in search of the goal without any additional cues. Table 1 summarizes the success rates for this task in indoor and outdoor environments. We find that, while end-to-end baselines avoid collisions with their surroundings, they fail to explore new areas and often get stuck in small areas of the environment. Graph-based methods avoid this trap by explicitly reasoning about coverage with the search objective, leading to a high success rate for ViNT. Qualitative analysis (Table 1-*right*) shows that planning over the diverse subgoals proposed using diffusion leads to more efficient paths, whereas other baselines take winding paths while exploring. Figure 11 illustrates the egocentric rollouts of the coverage exploration task in challenging indoor environments. ViNT-R performs respectably despite the lack of valid subgoal proposals.

This observation extends to the position-guided navigation task (Table 9), where the robots are tasked with reaching a *2D goal position* in a previously unseen environment. The robots have access to onboard wheel odometry (indoors), GPS coordinates (outdoors), or passive satellite imagery (outdoors), to track their position and use as a goal-directed heuristic. Compared to a baseline of the previous state of the art [29], we find that the various sub-goal predictions from the diffusion model

paired with the graph-based scoring scheme lead to a higher success rate and a greater distance traveled without collisions. ViNT is also more effective at avoiding collisions in crowded indoor spaces, and more efficient at reaching goals in outdoor spaces (captured by the SPL metric), owing to the implicit affordances and preferences learned by the large-scale pre-training (see further analysis in Section F. ViNT also requires fewer interventions, observed by the larger distance before observed collisions. Figure 12 illustrates a rollout of physical search in an outdoor environment with ViNT using satellite image as context (also see Figure 11).

## 6.2 Zero-Shot Generalization: a Single Policy to Drive Any Robot

Towards answering **Q2**, we deploy the *same* pre-trained ViNT policy on four distinct robotic platforms *without* any fine-tuning for the task of undirected exploration. We report the maximum displacement of the robot (in meters) from its starting position, without interventions, as a proxy for reaching arbitrarily distant goals in complex environments in Table 2. Most notably, ViNT successfully generalizes zero-shot to control a Go 1 quadruped, which does not appear during training.

We compare ViNT trained across all the combined datasets and robots to the best single-robot baseline — a model trained using data only from the target environment — as well as the GNM model [19] trained on all datasets. We observe that policies trained across robot embodiments can not only match, but also *outperform*, single-robot models across all the embodiments we studied. We also find that the larger capacity of ViNT leads to improved generalization compared to the smaller GNM model, especially on robots that do not appear in the training dataset (e.g., Go 1). Crucially, we also find that ViNT demonstrates *positive transfer* for in-domain robots (Vizbot), greatly outperforming a specialist model trained on only the target robot and setting, an emergent phenomenon not present in smaller models. This indicates that the model generalizes between tasks to improve performance, a key property of a foundation model.

| Model | LoCoBot | Go 1 | Vizbot | Jackal |
|---|---|---|---|---|
| Single-Robot | 40 | 12 | 40 | 184 |
| GNM [19] | 60 | 8 | 20 | **427** |
| ViNT | **120** | **45** | **110** | **438** |

**Table 2:** In coverage tasks, ViNT drives different robots for 100s of meters (reported maximum displacement without intervention), beating lower-capacity models (GNM) and specialist models trained on a single robot dataset.

## 6.3 Broader Generalization via Fine-Tuning

To answer **Q3**, we consider the problem of fine-tuning ViNT in the low data regime. In this setting, the entire ViNT model is fine-tuned end-to-end with a reduced learning rate of $1 \times 10^{-4}$ over $n_{ep} = 5$ epochs (Section 5). We assume access to a small amount of on-task data (at most 5 hours, with successful results in 1-2 hours of data), and study the the efficiency of learning from subsets of this data using ViNT. We study fine-tuning for the task of autonomous driving in the CARLA simulator for two reasons: (i) the simulated CARLA environment is perceptually distinct from the real-world data used to train ViNT (Fig. 13), and (ii) the on-road driving task requires very specific semantic behavior, i.e., driving in a lane and making smooth turns, that is not present in our real-world training data. We show that ViNT can be fine-tuned on a small amount of data (under 1 hour) to achieve strong performance by effectively leveraging the navigational priors encoded in the model.

Table 3 summarizes our findings. We report fractional progress towards the goal as "success", and the fraction of trajectories where the agent drives within the driving lane as "in lane". While pre-trained visual representations significantly improve task performance over a policy trained entirely from scratch, we observe that the learned policies suffer from frequent collisions and poor performance. GNM [19] outperforms these baselines due to its strong navigation prior, but the lower-capacity model is unable to generalize fully to the task. ViNT, on the other hand, is able to achieve strong performance, achieving substantially higher success rate than other baselines. Sweeping over fine-tuning dataset size (Table 3-right) shows that ViNT achieves strong performance with under 1 hour of fine-tuning data, demonstrating its ability to efficiently generalize to new environments.

| Method | Images | | Positions | Routing |
| | Success | In Lane | Success | Success |
|---|---|---|---|---|
| Scratch | 0.45 | 0.74 | 0.79 | 0.43 |
| ImageNet | 0.22 | 0.71 | 0.59 | 0.45 |
| SimCLR [7] | 0.21 | 0.63 | 0.70 | 0.64 |
| VC-1 [41] | 0.19 | 0.65 | 0.49 | 0.38 |
| GNM [19] | 0.49 | 0.66 | 0.45 | 0.49 |
| ViNT | **0.82** | **0.82** | **0.89** | **0.72** |

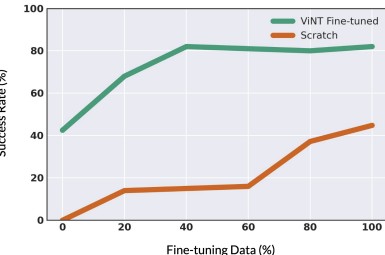

**Table 3:** *Left:* ViNT can be fine-tuned end-to-end (**Images**) or adapted to downstream tasks (**Positions** and **Routing**), and outperforms training from scratch and other pre-training methods. *Right:* ViNT can transfer navigational affordances to novel tasks (40% success without fine-tuning), and efficiently masters the task (80% success) with less than 1 hour of fine-tuning data.

## 6.4 Adapting ViNT to Downstream Tasks

To evaluate **Q4**, we investigate whether ViNT can serve as a foundation model for a broader range of downstream tasks by considering goal modalities beyond subgoal images (see Section 6.4). We consider the same CARLA driving task but with two different high-level planners: (i) a position-based planner that commands a sequence of GPS waypoints, and (ii) a routing planner with similar functionality to Google Maps that commands high-level navigation directions (left/right/straight) to the policy [22]. We compare the pre-trained navigational priors learned by ViNT to the baselines discussed earlier, corresponding to pre-trained visual representations and policies, each adapted to the downstream task using the same on-task data (see Appendix E.3 for more details).

Table 3 summarizes our results for the two tasks. We again find that general pre-trained visual representations, such as ImageNet or VC-1, are not sufficient to extract navigational affordances for challenging downstream tasks, suggesting that effective generalization requires more than general visual representations [41, 42]. We also find that unlike fine-tuning, GNM struggles with the adaptation task, suggesting that the architecture and increased capacity of ViNT are essential for broad generalization and adaptation.

## 7 Discussion

We presented ViNT, a robot foundation model that is trained for a generic image-goal navigation task on diverse data from many different robots, and can then support different navigation functionalities. ViNT can be deployed for long-horizon navigation in combination with a high-level planner, explore new environments with goals proposed by a diffusion model, be fine-tuned to new domains (such as autonomous driving), and be adapted to new task specification methods, such as turn-by-turn routing commands. Our results show that ViNT can successfully generalize across robots and environments, outperforms prior navigational models, can be efficiently fine-tuned to new domains and tasks, and shows promising emergent behaviors such as navigating through dynamic pedestrians.

**Limitations and Future Work**

As with many large-scale models, ViNT carries a heavier computational burden at inference time, which can present a challenge for power-constrained platforms such as quadcopters. While our design aims to enable efficient inference, our Transformer-based model is still significantly costlier to run at deployment time than simpler feedforward convolutional networks. Additionally, although ViNT generalizes effectively across robots in our experiments, it assumes a degree of structural similarity. More broadly, while ViNT illustrates the promise of a general-purpose and broadly reusable navigational foundation model, we believe that the most exciting developments for general-purpose cross-robot models are still ahead: as larger and larger multi-robot datasets are assembled, perhaps we will see even broader generalization and more flexible specification with increasingly powerful and general-purpose robotic models. We hope that ViNT represents a step in this direction.

## Acknowledgments

This research was partly supported by ARL DCIST CRA W911NF-17-2-0181, ARO W911NF-21-1-0097, IIS-2150826, and compute from Google TPU Research Cloud, NVIDIA, NSF CloudBank, and the Berkeley Research Computing program. The authors would like to thank Yunhao Cao, Seung-Hyun Kwon, Hrish Leen, Laura Smith, Medini Tolpadi, and Christopher Yu, for help with setting up experiments; Ted Xiao, for help with reproducing RT-1; Chethan Bhateja, Devendra Singh Chaplot, Kuan Fang, Adrien Gaidon, Dibya Ghosh, Saurabh Gupta, Haresh Karnan, Vijay Kumar, Hrish Leen, Fangchen Liu, Arjun Majumdar, Carlos Nieto, Aravind Rajeswaran, Ethan Stump, Jie Tan, Joanne Truong, and Tingnan Zhang, for useful discussions and feedback during various stages of this research. The authors would also like to thank Byron Boots, Gregory Kahn, Haresh Karnan, Xiangyun Meng, and Xuesu Xiao, for their help in aggregating the dataset used for training ViNT.

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

# Appendix

## A  ViNT Model Architecture

Table 4 shows the ViNT model architecture in detail. We use all 18 layers of the EfficientNet-B0 convolutional encoder [34], initialized from scratch, to tokenize the observation and subgoal images into 512-dimensional embeddings each. We utilize an observation encoder to tokenize the past and current observation images and a joint observation and goal encoder to tokenize the subgoal image fused with the current observation image channel-wise. For tokenizing the joint observation and subgoal token, images $o_t$ and $o_s$ are concatenated along their channel dimension, yielding a $6 \times 85 \times 64$ tensor per training data point.

| Layer | Input [Dimensions] | Output [Dimensions] | Layer Details |
|---|---|---|---|
| 1 | $o_t$, $o_s$ [64, 85, 3] | $I_t^g$ [64, 85, 6] | Concatenate observations and goal |
| 2 | $I_t^s$ [64, 85, 6] | $E_t^s$ [1, 1000] | Goal EfficientNet encoder |
| 3 | $o_{t:t-P}$ [P+1, 64, 85, 3] | $E_{t:t-P}$ [P+1, 1000] | Context EfficientNet encoder |
| 4 | $E_t^s$ [1, 1000] | $E_t^{s'}$ [1, 512] | Goal embedding compression |
| 5 | $E_{t:t-P}$ [P+1, 1000] | $E_{t:t-P}'$ [P+1, 512] | Context embedding compression |
| 6 | $E_{t:t-P}'$ [P+1, 512], $E_t^{s'}$ [1, 512] | $S$ [P+2, 512] | Concatenate |
| 7 | $S$ [P+2, 512] | $\tilde{S}$ [1, 32] | Feed into Transformer $f$ |
| 8 | $\tilde{S}$ [1, 32] | $d$ | Predict temporal distance $d$ |
| 9 | $\tilde{S}$ [1, 32] | $\hat{a}$, [1, T, 4] | Predict future actions $\hat{a}$ |

**Table 4: Architectural Details of ViNT** The inputs to the model are RGB images $o_{t:t-P} \in [0, 1]^{P \times 3 \times 85 \times 64}$ and $o_s \in [0, 1]^{3 \times 85 \times 64}$, representing the current, past, and goal images. We seek to predict a $H$ future actions $\hat{a}$ and the temporal distance $d$.

### A.1  Goal-Conditioning Architectures

We considered different mechanisms for conditioning ViNT with features from the subgoal image, as illustrated in Figure 5.

1. **Late Fusion:** Extract the observation and goal features independently and fuse them in the multi-head attention layers. To achieve this effect, we avoid any channel-wise concatenation between any of the observation and goal images before inputting them into the model.

2. **Early Fusion:** Jointly extract observation (context) and goal features and fuse the observation and goal features before we tokenize them. We achieve this by concatenating the goal image with every observation image along the channel dimension. We remove the goal token in this setup since information about the goal is already in every observation token.

3. **FiLM (RT-1):** Following the FiLM+EfficientNet encoder (Brohan et al. [30]), encode each observation image separately. For conditioning on visual goals, we replace the "Universal Sentence Encoder" with an EfficientNet encoder. We remove the goal token in this setup since information about the goal is already in every observation token.

Our observations are summarized in Table 5. While FiLM works well for language, we found that training was unstable for image-based navigation tasks. Instead, we directly encode each observation independently and pass them to a Transformer. Ideally, the goal would be encoded separately and then combined with the observations in the Transformer layers, allowing the entire goal encoder to later be swapped out for different goal modalities. Unfortunately, we found that this approach (which we term "late fusion", as the goal and observations are not fused until *after* encoding them) performs poorly: in image-based navigation, it is the *relative* features between the observation and

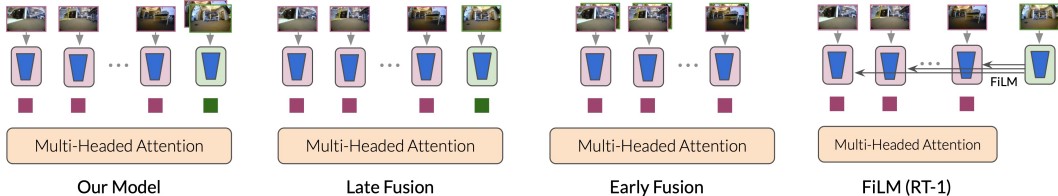

**Figure 5:** Different goal-conditioning architectures considered for ViNT.

| Method | Performance | Adaptation |
|---|---|---|
| Late Fusion | ✗ | ✓ |
| Early Fusion | ✓ | ✗ |
| FiLM (RT-1) [30] | ✗ | ✓ |
| ViNT | ✓ | ✓ |

**Table 5:** Comparing merits (✓) and demerits (✗) of different goal-conditioning architectures. While "Early Fusion" works the best for the core navigation task, it does not support downstream adaptation (Section 5). "Late Fusion" is ideal for adaptation, but does not perform well for our tasks. Our goal fusion architecture is able to closely match the performance of early fusion, while also supporting adaptation.

goal images that are important, rather than *absolute* goal features. An "early fusion" architecture would fuse the goal image with all the past and current observation images immediately, which allows for learning joint features between the goal image and current state. However, this architecture is inflexible as the observation encoder would have to be learned entirely from scratch when adapting to a new goal modality. ViNT avoids this issue by using two distinct types of encoders: an observation-only encoder used to tokenize each observation image, and a joint observation and goal encoder that should extract relative goal features. This latter encoder can be replaced to allow alternative goal specifications in downstream tasks, as described in Appendix B.4. Specifically, we adapt to new tasks by learning the final token conditioned on the new task goal information in place of the joint observation/goal encoder.

## B    Implementation Details

### B.1    Training ViNT

See Table 6 for a detailed list of hyperparameters for training the ViNT foundation model.[1]

### B.2    Subgoal Diffusion

For generating subgoals, we use an image-to-image diffusion model. It takes an image $o_t$ as input and produces samples from $g(o_{s_i} \mid o_t)$, where $o_{s_i}$ are candidate subgoal images reachable from $o_t$. To produce training pairs for the diffusion model, we first select $o_t$ uniformly at random from the training data and then select $o_{s_i}$ to fall between 5 and 20 timesteps in the future from $o_t$.

Following Saharia et al. [36], we implement image conditioning as simple channel-wise concatenation to the U-Net input. We use the Flax U-Net implementation from the diffusers library [45] with textual cross-attention removed since we do not condition on text inputs.

We use the continuous-time diffusion formulation from Kingma et al. [46] with a fixed linear noise schedule rather than a learned one. Also unlike Kingma et al. [46], we use the unweighted training objective, called $L_{\text{simple}}$ in Ho et al. [35, Equation 14] and Kingma et al. [46, Appendix K]. We

---

[1]We used a variety of workstations equipped with different GPU configurations over the course of this research, including 2×4090, 3×Titan Xp, 4×P100, 8×1080Ti, 8×V100, and 8×A100. With the model architecture fixed, the batch size and training time varies significantly across these devices, and the entry in Table 6 is representative of our most common training configuration.

| Hyperparameter | Value | Hyperparameter | Value |
|---|---|---|---|
| **ViNT Model** | | **Diffusion Training** | |
| # Parameters | 31M | Dropout | 0.1 |
| RGB Resolution | $85 \times 64$ | Batch Size | 128 |
| Encoder | EfficientNet-B0 | Optimizer | AdamW |
| Token Dimension | 512 | Warmup Steps | 1000 |
| Attn. hidden dim. | 2048 | Learning Rate | 1e-4 |
| # Attention Layers $n_L$ | 4 | LR Schedule | Cosine |
| # Attention Heads $n_H$ | 4 | Adam $\beta_1$ | 0.95 |
| Temporal Context $P$ | 5 | Adam $\beta_2$ | 0.999 |
| Prediction Horizon $H$ | 5 | Adam $\epsilon$ | 1e-8 |
| MLP layers | (256, 128, 64, 32) | Weight Decay | 0.001 |
| **ViNT Training** | | EMA Inv. Gamma | 1.0 |
| # Epochs $n_{\text{ep}}$ | 30 | EMA Power | 0.75 |
| Batch Size | 300 [1] | EMA Max Decay | 0.9999 |
| Learning Rate | $5 \times 10^{-4}$ | CFG Mask Proportion | 0.2 |
| Optimizer | AdamW [43] | Train Steps | 250,000 |
| Warmup Epochs | 4 | Training Time | 30 hours |
| LR Schedule | Cosine | Compute Resources | v4-8 TPU board |
| Scheduler Period | 10 | **Diffusion Sampling** | |
| Compute Resources | $8 \times$ V100 [1] | Sampler | DDIM [44] |
| Training Time | 30 hours [1] | DDIM $\eta$ | 0.0 |
| Fine-tuning LR | $1 \times 10^{-4}$ | Sampling Steps | 200 |
| **Diffusion Model** | | Guidance Weight | 1.0 |
| # Parameters | 318M | **Other** | |
| Resolution | $128 \times 128$ | Maximum distance | 20 |
| # Up/Down Blocks | 4 | Distance tradeoff $\lambda$ | 0.01 |
| Attn. Resolutions | 32, 16, 8 | | |
| Layers per Block | 2 | | |
| Attn. Head Dim | 8 | | |
| Channels | (128, 128, 256, 512, 640) | | |
| Diffusion Type | continuous time | | |
| Noise Schedule | linear | | |

**Table 6:** Hyperparameters for training ViNT and the diffusion model.

employ classifier-free guidance [47] and find that it helps produce subgoals with better visual fidelity, which is consistent with prior work [48].

### B.3 Long-Horizon Physical Search via Topological Graphs

As in Shah and Levine [29], we implement physical search similarly to a standard A* algorithm, by keeping track of an open set $\Omega$ of possible unvisited subgoals (generated by our diffusion model) and following Alg. 1.

Nodes are visited according to a costing function $f(s)$ that depends on the distance from the current state $o_t$ to the parent node $s^-$ (measured along the graph), the predicted distance from $s^-$ to $s$, and a heuristic function $h$ (similar to that of A*) providing long-horizon navigation hints:

$$f(s) = d_{\mathcal{M}}(o_t, s^-) + d_{\text{pred}}(s^-, s) + h(s, G, C)$$

In general, the heuristic can be any function providing a notion of distance between a subgoal $s$ and the long-horizon goal $G$, optionally with some context $C$. For our experiments, we considered three heuristics to demonstrate the flexibility of our approach:

- **Coverage exploration:** We have no long-horizon guidance for coverage exploration, and thus, use $h(s) = 0$.

- **Position-guided:** For long-horizon GPS goals (outdoors) and 2D position goals (indoors), we use Euclidean distance $h(s) = \|s - G\|$.

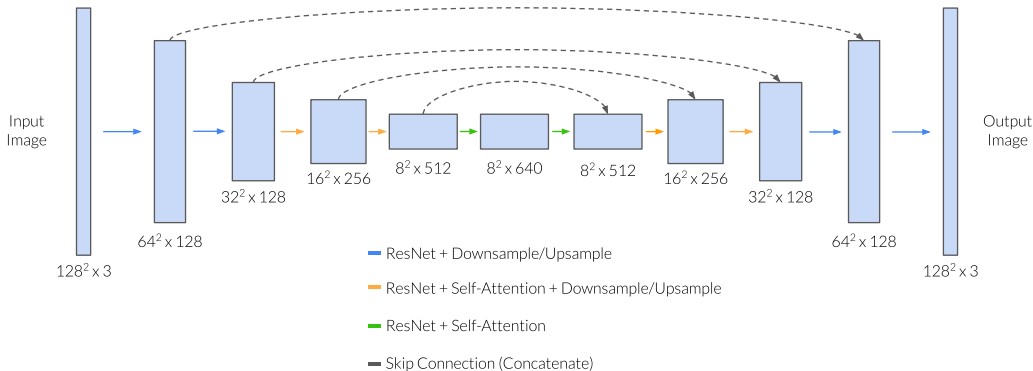

**Figure 6:** Subgoal diffusion model U-Net architecture. Each ResNet consists of 2 residual blocks. Downsampling and upsampling is done with strided convolutions.

---

**Algorithm 1:** Long-Horizon Navigation via Topological Graph

---
1: **while** goal $G$ not reached **do**
2:     $s \leftarrow \min_f(\Omega)$;
3:     $P \leftarrow \text{ShortestPath}(\mathcal{M}, o_t, s^-)$
4:     **for** $(s, s')$ in $P$ **do**
5:         ViNT.GoToGoal($s'$);
6:     **end for**
7:     ViNT.GoToGoal($s$)
8:     $o_t \leftarrow \text{Observe}()$;
9:     AddNode($\mathcal{M}, o_t$, parent: $s^-$);
10:     Sample $s_i \sim g(s_i | o_t)$;
11:     Add($\Omega, s_i$);
12: **end while**

---

- **Satellite-guided:** In the context-guided experiments, we train a learned heuristic function that uses the satellite image as an input to learn a a heuristic for "good" subgoals. We train a convolutional neural network on the overhead image to predict the probability that the subgoal $s$ is included on a trajectory from $o_t$ to $G$, trained using a contrastive objective [49]. Additional information can be found in Shah and Levine [29].

## B.4   Fine-tuning ViNT

In all CARLA fine-tuning experiments, on-task data was collected using a rule-based oracle agent, with start and end locations sampled randomly up to 900 meters apart. We collect 181 training trajectories (roughly 4 hours) in CARLA's Town 01 environment, and a further 52 trajectories (1 hour) in the held-out Town 02 environment. Inspired by Codevilla et al. [22], we further augment this dataset by allowing the rule-based agent to correct its position and re-center to the lane after a perturbation.

**Image Fine-tuning:**

- **Architecture:** We utilize the exact same architecture as ViNT with no changes.

- **Training:** For fine-tuning the image-goal directed model, we utilize the same training process for ViNT with a learning rate of 0.0001, AdamW optimizer, but no warmup or cosine scheduler. We do not mix any prior data for fine-tuned training.

**GPS-Adaptation:**

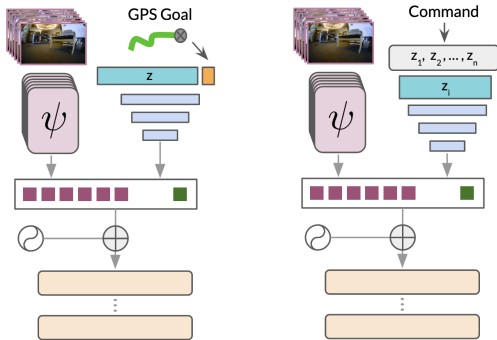

**Figure 7:** Adaptation architectures for ViNT. Left: GPS-adaptation architecture. The local coordinates of the goal are concatenated to the fixed latent $z$. Right: command-adaptation architecture, using latent $z_i$ selected by command label index $i$.

- **Architecture:** To adapt to GPS-style goals, we cut off the goal encoder block from ViNT. We then learn a fixed tensor of size 3000 and concatenate it to the GPS-command goal in ego-centric coordinates. We then pass this into a 2-layer MLP which outputs the prediction of the final token for the transformer. The architecture is shown in Figure 7.
- **Training:** During training, instead of randomly sampling future images to serve as goals, we sample goals from future odometry information. Once we have a future goal coordinate for self-supervision, we convert to local coordinates and pass into our architecture, fine-tuning with the same objective as ViNT. We use a cosine scheduler with a learning rate warmup to 0.0001 for 4 epochs. We also sample goal points from between 1.25s and 1.75s rather than from 0.5s to 2.5s.

**Command-Adaptation:**

- **Architecture:** For discrete command goals, we adopt a similar approach for GPS-style goals. We learn a fixed tensor for each discrete command and use the command index to select the corresponding latent to pass into a 2-layer MLP for predicting the final token. In this way, we learn a dictionary of latents, each corresponding to a distinct command. This architecture is illustrated in Figure 7.
- **Training:** For our experiments, we use "left", "right", and "straight" as our discrete commands. We assume training data is not labelled with the discrete command, so we label dataset trajectories with the corresponding commands retroactively by sampling a future position (as in GPS-Adaptation) and then selecting a command based on its lateral deviation. For our experiments we bin samples with lateral coordinate greater than 0.05 as "left" or "right" and label the remaining samples as "straight". We again use a cosine scheduler with a learning rate warmup to 0.0001 for 4 epochs.

## C    Training Dataset

The ViNT training dataset contains over 100 hours of real-world navigation trajectories, sourced entirely from existing datasets. The dataset consists of a combination of tele-operated and autonomous navigation behaviors collected across 8 distinct robotic platforms, including 4 commercially available platforms (TurtleBot, Clearpath Jackal, Warthog and Spot) and several custom platforms (Yamaha Viking ATV, RC Car, passenger automobiles). The trajectories contain widely varying robot dynamics and top speeds, ranging between 0.2 and 10m/s, operating in a diverse set of environments (e.g., office buildings, hallways, suburban, off-road trails, university campuses, etc.). All data is either publicly available, or collected by *other* researchers for past projects; *no additional training data was collected specifically for training ViNT.*

|    | Dataset | Platform | Speed | Total Hrs. | Hrs. Used | Environment |
|----|---------|----------|-------|-----------|-----------|-------------|
| 1  | GoStanford [28] | TurtleBot2 | 0.5m/s | 17h | 14h | office |
| 2  | RECON [40] | Jackal | 1m/s | 25h | 25h | off-road |
| 3  | CoryHall [39] | RC Car | 1.2m/s | 2h | 2h | hallways |
| 4  | Berkeley [29] | Jackal | 2m/s | 4h | 4h | suburban |
| 5  | SCAND-S [50] | Spot | 1.5m/s | 8h | 4h | sidewalks |
| 6  | SCAND-J [50] | Jackal | 2m/s | 1h | 1h | sidewalks |
| 7  | Seattle [51] | Warthog | 5m/s | 1h | 1h | off-road |
| 8  | TartanDrive [52] | ATV | 10m/s | 7h | 5h | off-road |
| 9  | NeBula [53] | ATV | 10m/s | 10h | 10h | off-road |
| 10 | SACSoN [54] | TurtleBot2 | 0.5m/s | 75h | 10h | office |
| 11 | BDD [13] | Car(s) | 20m/s | 10h | 4h | on-road |
|    | Total |  |  | 160h | 80h |  |

**Table 7: The ViNT training dataset** contains over 150 hours of navigation data in challenging indoor, outdoor, and off-road environments across 8 different robots of varying sizes, speeds, and capabilities.

Remember to mention: total size, number of robots, conversion to number of frames and so on.

## D   Robotic Platforms for Evaluating ViNT

**Vizbot:** A custom-built robot platform inspired by the design of Niwa et al. [55], based on a Roomba. It is equipped with an off-the-shelf PCB-mounted fisheye camera.

**Unitree Go 1:** A commercially available quadruped robot equipped with the original forward-facing camera. *There is no training data from a Go 1 in the training dataset.* Athough SCAND includes data collected on a Boston Dynamics Spot, which is also a quadrupedal robot, the two platforms practically have very different characteristics.

**Clearpath Jackal UGV:** A commercially available off-road platform equipped with an off-the-shelf PCB-mounted fisheye camera. *This system resembles the data collection platform used for the RECON, Berkeley, and SCAND-J datasets*, but has a different camera and mounting height.

**LoCoBot:** A popular open-source platform based on a Kobuki equipped with an off-the-shelf PCB-mounted fisheye camera. *This robot is not present in the training dataset*, although GS was collected on a similar TurtleBot2 with a different spherical camera at a lower height.

## E   Evaluation Setup and Details

### E.1   Navigation Performance

#### E.1.1   Indoor Experiments

For setting up the indoor coverage exploration experiments, we use the LoCoBot and Vizbot robotic platforms. We choose a random starting point and goal in an enclosed environment, and keep these locations consistent across all baselines we test. For the coverage exploration task, we ensure that the environments are "enclosed" and block any glass walls and stairwells, which are beyond the capabilities of the robots. Experiments are terminated when either (i) the robot is unable to reach the goal within a pre-specified time limit of 10 minutes, or (ii) the robot becomes physically stuck (e.g., collides and is unable to recover).

For setting up the indoor guidance exploration experiments on the LoCoBot, we mark the start and goal locations in a large office building and note their 2D positions. The goal location is conveyed to the robot as the context, and is available to the search algorithm. The system uses the robot's onboard wheel odometry to track position.

### E.1.2 Outdoor Experiments

For the coverage exploration experiments, we follow the setup of Shah et al. [40] and use the Clearpath Jackal UGV. We choose a random start and goal location in confined outdoor environments and obtain a goal image observation for the robot to seek. Experiments are terminated either when (i) the robot is unable to reach the goal within a pre-specified time limit of 20 minutes, or (ii) the robot collides with an obstacle in the environment.

For the guided exploration experiments, we closely follow the setup of Shah and Levine [29]. For the GPS guided experiments, the robot has access to the GPS location of the goal, in addition to a goal image. For the satellite-guided experments, the robot further has access to an overhead satellite image centered at its current location and a learned heuristic funtion $h$.

### E.1.3 Baselines

For experiments presented in Section 6.1, we evaluate 4 baselines against our method.

1. **End-to-End BC:** A modified ViNT model with no goal token, trained end-to-end for the task of only predicting future actions. This represents a typical undirected BC baseline with similar model capacity as the other baselines.

2. **End-to-End GCG:** A model-based algorithm that uses a predictive model to plan a sequence of actions that reach the goal without causing collisions [39]. Since this model requires collision labels for training, it is only trained on a subset of the training data (RECON, CoryHall, Berkeley) that has these labels; hence, this baseline is only evaluated outdoors.

3. **RECON:** A variant of the physical search algorithm RECON [40], which uses a latent goal model to represent reachable goals and plans over sampled subgoals to explore the environment in a similar manner to ours. This baseline uses a variational information bottleneck to sample latent subgoals, rather than a diffusion model sampling subgoal images.

4. **ViNT-R:** An ablation of our method that uses subgoals randomly sampled from the training data, instead of samples from a conditional diffusion model, as subgoal candidates.

## E.2 Multi-robot Generalization Experiments

The setup for the multi-robot generalization experiment is same as the coverage exploration experiments. The only differences are the baselines we evaluate.

### E.2.1 Baselines

For experiments presented in Section 6.2, we test three baseline low-level policies on each robot. Each baseline uses the graph-based exploration scheme described in Section 4.1. We use the following baselines:

1. **Single-Robot:** We train a single-dataset policy model (ViNT architecture) and diffusion model on the two largest datasets (RECON for outdoor, and SACSoN for indoor), and evaluate them on each of our robots to identify the best single-dataset model for each robot. Note that we do not have comparable magnitudes of training data of visual locomotion on the Go 1.

2. **GNM:** We use the pre-trained model checkpoint from the authors of GNM [19] coupled with *our* diffusion model (since GNM is not compatible with the exploration task) to evaluate each robot.

3. **ViNT:** We use our pre-trained ViNT policy and image diffusion model (no fine-tuning) to evaluate each robot.

### E.3   Fine-tuning and Adaptation

This section describes the setup and implementation details for ViNT fine-tuning and adaptation experiments in the CARLA autonomous driving simulator, as presented in Sections 6.3 and 6.4.

### E.3.1   CARLA Data Collection

We collect expert trajectories with an oracle rule-based self-driving agent and gather odometry and RGB information across the trajectory at 4 Hz. These trajectories have random spawn points and random destination points up to to 900 meters in length. We collect 52 trajectories in the CARLA Town 02 for held-out testing, and collect 181 trajectories in Town 01 for training. This makes for a dataset size of 5 hours for the autopilot control data. Inspired by [22], we also collect short trajectories of the agent correcting back onto the right lane after drifting off course in Town 01 and Town 02 for training and testing, respectively. This data is 4 hours long, and we add it to the autopilot data for training.

### E.3.2   Fine-tuning Experiments

To test the fine-tuning system which trains ViNT with the same goal specification but in new domains, we utilize the collected test trajectories as sequences of goal images to follow. Each Town 02 test trajectory creates a graph in which every node is a timestamped odometry point corresponding to an image. To evaluate a model on a test trajectory, we spawn it at the same start point and localize it on the trajectory's map. We then query the image for the goal node which corresponds to node 1.5s after the current node. This goal image is sent to ViNT along with the 4Hz image context to compute a short-range trajectory. This is tracked by a simple PID controller. The average progress towards the goal before collision is collected and reported across all trials. Table 3 summarizes the results of these experiments with multiple baselines and data sizes.

### E.3.3   Adaptation Experiments

To test the new tasks, we adopt a similar evaluation setup to the fine-tuning experiments, but rely on the odometry position for the selected goal node rather than the image. For positional-adaptation, we move the goal coordinates into a local frame and send it to ViNT. For routing-adaptation, we determine the difference in lateral coordinates between the current node and the goal node. We choose the current node as reference to ensure an open-loop experiment and to allow for pre-computation of the command signals to be sent. We then apply the same binning strategy during training using a 0.05 normalized distance as the boundary between "left", "right", and "straight". The control system downstream of this is identical to image fine-tuning and the experiment terminates when at the goal or when colliding. The progress towards the goal before collision is collected and averaged across all trials in Table 3.

### E.3.4   Baselines

We have the following baselines for the CARLA experiments:

1. **Scratch**: ViNT trained from scratch on the CARLA on-task dataset.

2. **Pre-trained Visual Representations**
   (a) **ImageNet**: ViNT initialized with the EfficientNet-B0 weights pre-trained on ImageNet, other parameters initialized from scratch, and fine-tuned with the CARLA on-task dataset.
   (b) **SimCLR**: ViNT initialized with the EfficientNet-B0 weights pre-trained with Sim-CLR [7] on the training data described in Section C, other parameters initialized from scratch, and fine-tuned with the CARLA on-task dataset.
   (c) **VC-1**: ViNT initialized with a pre-trained ViT-B model checkpoint from the authors of VC-1 [41] *and frozen*, other parameters initialized from scratch, and fine-tuned

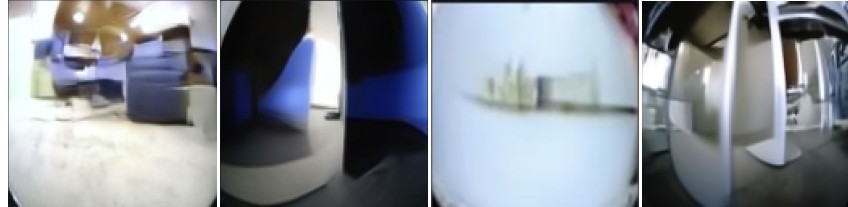

**Figure 8: Samples from the diffusion model** may be invalid subgoals, but ViNT is robust to such proposals.

with the CARLA on-task dataset. The VC-1 encoder is pre-trained on a combination of Ego4D, manipulation, navigation, and ImageNet images using Masked Auto-Encoding [56, 57].

3. **GNM**: The pre-trained embodiment-agnostic model checkpoint from the authors of GNM [19], fine-tuned with the CARLA on-task dataset. Note that GNM has 8.7M trainable parameters, compared to ViNT's 31M.

We note that the VC-1 baseline's weak performance in Section 6.4 may be explained by the fact that it is *frozen*, while all other visual encoders were free to fine-tune. This is representative of typical downstream usage [41]. Despite training on multiple, diverse datasets, the visual representation's general-purpose features are not optimized for the navigation task, hampering zero-shot transfer to out-of-domain tasks. To provide a fair comparison of the quality of pre-trained visual features, we compare this performance to ViNT-FE (a pre-trained

| Method | Images | Positions |
|---|---|---|
| VC-1 [41] | 0.19 | 0.49 |
| ViNT-FE | 0.32 | 0.78 |
| ViNT | 0.82 | 0.89 |

**Table 8:** Evaluation of ViNT fine-tuning with and without a frozen encoder, as compared to a general-purpose visual encoder. Even when frozen, ViNT's navigation-relevant features appear to transfer more readily to out-of-distribution inputs than general-purpose features.

ViNT model that has it's visual encoder frozen). ViNT-FE has an equal number of trainable parameters to the VC-1 baseline, and frozen visual representations (see Table 8).

## F    Emergent Behaviors

One of the most exciting aspects of large-scale machine learning is the potential for emergent behavior that arises from the training of large models on diverse datasets. Despite the simple self-supervised training objective used by ViNT, it shows a number of emergent behaviors, which we describe qualitatively in this section and present as examples on the project page and supplemental videos: general-navigation-models.github.io.

**Implicit navigation affordances:** Ideally, we would like a robot foundation model to exhibit some desirable "default" behavior, while providing a mechanism for downstream applications to adapt this behavior as needed. We find that ViNT has this property vis-a-vis collision-avoidance. One piece of evidence is its behavior when provided with *random* subgoals from locations that are not reachable by the robot, studied quantatively via the ViNT-R baseline in Table 1. In this case, despite the subgoals being invalid and out-of-distribution (ViNT was only trained to reach subgoals), ViNT succeeds at exploring the environment and reaches the goal 80% of the time, outperforming all baselines. This suggests that ViNT takes collision-free actions when provided with meaningless goals (i.e. the above "default"), while still attempting to follow reachable subgoals.

Indeed, although the "full" version of our method augmented with the diffusion model performs better, the subgoals generated by this model are often of low quality with many artifacts, and sometimes do not match any real reachable state (Figure 8). Nonetheless, because of this "default" behavior, ViNT is able to successfully leverage the valid subgoals, while ignoring the bad ones, and demonstrate collision-free navigation in previously unseen environments.

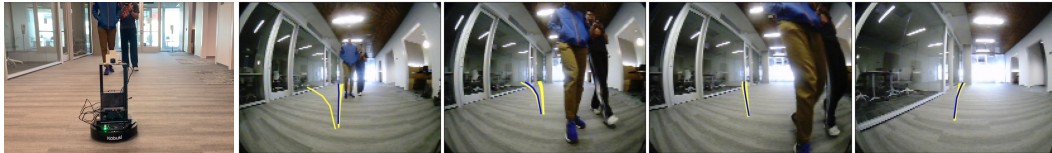

**Figure 10: Robustness to dynamic pedestrians.** ViNT can successfully navigate around a crowd of dynamic pedestrians and reach the goal behind them, despite its simple self-supervised training objective.

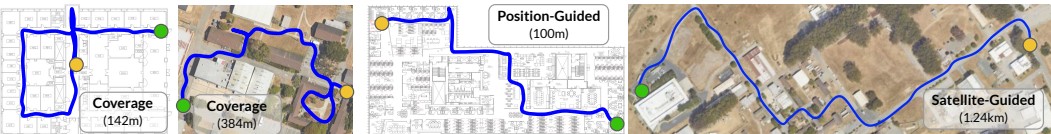

**Figure 11:** ViNT accomplishes long-horizon navigation with a variety of objectives in indoor and outdoor environments; example trajectories between start (orange) and goal (green) visualized here. Goal-reaching behavior can be achieved with a goal-directed heuristic (optionally guided by satellite imagery), while removing this heuristic allows for undirected exploration to maximally cover a workspace.

**Implicit navigation preferences:** Yet another interesting property exhibited by ViNT is its implicit preference to follow paved roads (outdoors), and drive smoothly in the middle of hallways (indoors), as demonstrated in Figure 9 and in the supplemental video. This is particularly interesting since a large chunk of the pre-training dataset contains suboptimal, weavy trajectories, and suggests that ViNT can learn "good" default behavior from the diverse train-

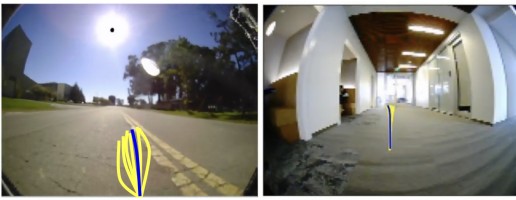

**Figure 9:** ViNT exhibits an implicit preference for following paved roads (*left*) and hallways (*right*).

ing behaviors. This preference helps ViNT efficiently explore previously unseen environments, where other baselines tend to explore the environment haphazardly (see Table 1 (*right*)).

**Robustness to dynamic pedestrians:** While ViNT is trained only on offline data with a simple, self-supervised training objective, we find that its collision avoidance capabilities generalize to dynamic obstacles and pedestrians. Figure 10 exhibits an instance where the robot is tasked with navigating to a goal behind two pedestrians. ViNT selects actions that avoid the pedestrians and recovers to the original path, successfully reaching the goal.

## G   Additional Results

Please see Table 9, and Figures 11 and 12 for more experiment rollouts.

| Method | Indoor: Position | | Outdoor: GPS | | | Outdoor: Satellite | | |
|---|---|---|---|---|---|---|---|---|
| | Success | Distance | Success | SPL | Distance | Success | SPL | Distance |
| ViKiNG [29] | 0.60 | 56m | 0.64 | 0.42 | 720m | 0.77 | 0.68 | 780m |
| ViNT | **0.90** | **91m** | **0.95** | **0.84** | **1270m** | **1.00** | **0.94** | **1040m** |

**Table 9:** ViNT can effectively utilize goal-directed heuristics, such as 2D goal positions and satellite images, to explore novel kilometer-scale environments successfully and without interventions.

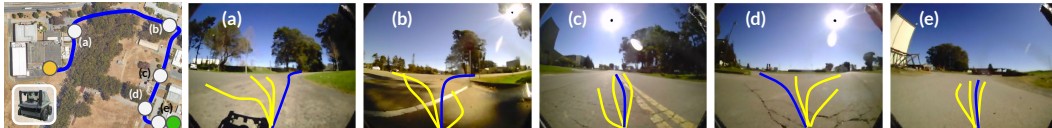

**Figure 12: Satellite-guided physical search with ViNT.** We visualize a 765m rollout of ViNT with a satellite image-based heuristic from start (orange) to goal (green). The future action samples $\hat{a}$ obtained by *spatially grounding* the subgoal candidates for five instances in the trajectory are shown in yellow. An A*-like planner uses the heuristic to pick the best subgoal (corresponding $\hat{a}$ marked in blue), guiding the robot to the goal.

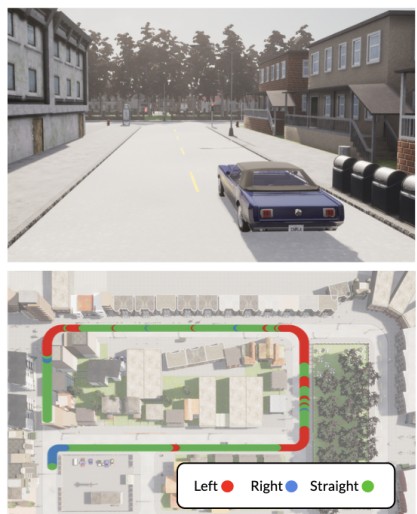

**Figure 13:** The CARLA test environment (*top*), and a bird's eye view showing high-level routing commands for the routing task.

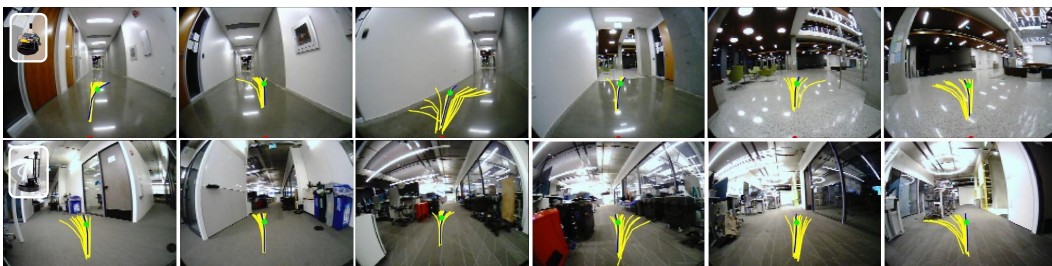

**Figure 14:** Visualizing ViNT exploration rollouts in challenging indoor environments using the Vizbot (top) and LoCoBot (bottom) robotic platforms. Future action samples $\hat{a}$ obtained by *spatially grounding* the subgoal candidates are shown in yellow, with the best actions corresponding to the best candidate marked in blue.

