# OpenReview forum: "ViNT: A Foundation Model for Visual Navigation"
_robot-learning.org/CoRL/2023/Conference — CoRL 2023 Oral_

### Official Review · Reviewer_Q7NJ · 2023-06-29

**Confidence:** 3
**Originality:** Very Good
**Technical Quality:** Excellent
**Clarity Of Presentation:** Excellent
**Impact:** 4

**Recommendation:**

Strong Accept: I recommend accepting the paper and will argue for my recommendation even if other reviewers hold a different opinion.

**Review:**

Strengths:
1. The motivation to build a foundation model is inspiring. Robot learning usually involves various tasks, platforms, and environments, which limits the available data for different scenarios. This paper can be a valuable attempt to generalist foundation models in robotics.
2. ViNT presents pre-training via predicting future actions and temporal distances. It demonstrates that such methods can be applied to heterogeneous real-world videos, which enlarge the available training data.
3. The authors implement a diffusion model to generate subgoals for long-horizon navigation which greatly broaden the application of the pretrained model.
4. The authors conduct rigorous experiments on different navigation settings and robotic platforms which demonstrate the effectiveness of the proposed method. Also, details regarding training and deployment are provided.
5. The paper is well-written and easy to follow.

Weaknesses:
1. The paper focuses on vision-based navigation. Though it shows adaptation to different modalities in some experiments, there are still other modalities that may not be plugged in directly.

**Quality Of The Limitations Section:**

Limitations are addressed clearly

**Questions For Rebuttal:**

Major questions:
1. How do authors determine to choose observations over the past 5 timesteps as input and future actions over 5 future timesteps? Will predicting more future actions be helpful?
2. When applying the diffusion model in generating downstream tasks, is it fine-tuned on downstream tasks? If not, will the diffusion model still generate plausible samples?
3. Long-horizon navigation relies on ViNT itself to grounding spatially. Will that lead to the accumulation of errors in cases where ViNT fails to accurately predict the temporal distances?
4. As shown in Table 4, pretraining strategies like ImageNet and SimCLR achieve worse performance than training scratch. What could be the cause of the results? Since these pretraining methods usually perform well on vision tasks.
4. As the authors also mention in the discussion, Transformer model can be computationally costly. I'm curious what the inference time of ViNT is compared with other baselines.

Minor questions:
1. In Appendix Table 7, FF dim of ViNT model is not correctly shown.
2. In Appendix section C, there seems to be a note sentence that should be deleted.

**Robotics Focus:**

Sufficient demonstration on hardware

**Summary Of Paper:**

In this work, the authors present Visual Navigation Transformer (ViNT), which aims at building a foundation model for vision-based navigation. ViNT tokenizes observations over the past 5 steps as well as the goal/observation states via EffcientNet-50. A Transformer is then applied to predict the 5 future actions and temporal distance to the goal. ViNT is pretrained on over 100 hrs heterogeneous navigation trajectories. The authors also propose implementation of diffusion model to generate subgoals and physical search for long-horizon planning. Through rigorous experiments, ViNT demonstrates (1) zero-shot generalization to unseen environments, (2) zero-shot generalization to different robotic platforms, (3) adaptation to low data regime via fine-tuning, and (4) adaptation to different modalities with different encoders.

**Summary Of Recommendation:**

As mentioned above, this paper attempts to build foundation models for vision-based navigations. The paper presents a Transformer-based architecture as well as a pretraining technique that can be applied to heterogeneous trajectories. It also applies a diffusion model to generate subgoals to enable long-horizon navigation. Rigorous experiments that demonstrate the generalizability and adaptability of the proposed ViNT in navigations under a wide variety of settings. Overall, this paper can be a valuable contribution to building foundation models for various robotic tasks.

---

### Official Review · Reviewer_LWck · 2023-07-17

**Confidence:** 3
**Originality:** Very Good
**Technical Quality:** Good
**Clarity Of Presentation:** Fair
**Impact:** 4

**Recommendation:**

Weak Reject: I recommend rejecting the paper, but will not argue for my recommendation if the majority of other reviewers have a different opinion.

**Review:**

Strengths
* It's a very exciting and trending idea to apply foundation models to robotics (and specifically robot navigation).
* Sec. 3 is written fairly clearly
* Questions asked in Sec. 6 are very useful questions to ask. It's particularly exciting that the proposed model can generalize to new environments and to new robots. It's also nice that it can be fine-tuned for out of distribution environments.
* Lots of experiments. The authors run a lot of experiments to answer several fundamental questions. Each of these experiments is valuable and answering relevant questions.

Weaknesses & Suggestions
* Please specify that the subgoals are represented as subgoal (RGB) images (line 147)
* Suggestions for related work which use topological maps for navigation, and also utilize waypoint prediction + low-level controller to reach navigation subgoals.
    * Chaplot et al. Neural Topological SLAM for Visual Navigation. CVPR 2020.
    * Chen et al. Topological Planning with Transformers for Vision-and-Language Navigation. CVPR 2021.
* Some important details are left out of the main paper and included in the appendix instead.
    * Line 173 skips over important details wrt the planner and includes them in the appendix instead. More details in the main paper would be appreciated.
    * Overall, I would appreciate more details in Sec. 4.2.
* Some important details are left out of the appendix (and main paper)
    * Line 556 seems to be added by accident: "Remember to mention: total size, number of robots, conversion to number of frames and so on."
        * Many details from the training dataset are missing, such as the above info. Were the training sets simply concatenated? What kind of processing was applied on the datasets and how was the data sampled (e.g. uniformly)?
    * What is d_M and d_pred in line 499 of appendix?
    * Where is f() defined? It seems to have two different definitions in the main paper vs in appendix.
    * Please include more details in Appendix B.4.
        * For example, it's unclear if ground truth odometry is used, and if there is a supervision signal for the goal encoder adapter \tilde{phi}().
        * What does it mean to "learn a fixed tensor" (lines 522, 533 of appendix)? Are you representing GPS commands as a latent learnable representation? How is this supervised?
* More qualitative results, especially for the baseline models as well, would be useful.
* Please clarify the output action space for all baselines (especially if they differ from the waypoints action space of the proposed model).
* The baselines are weak (e.g. end-to-end BC). Recommended baselines include:
    * Chaplot et al. Neural Topological SLAM for Visual Navigation. CVPR 2020.
    * Chaplot et al. Learning to Explore using Active Neural SLAM. ICLR 2020.
    * Chen et al. Learning Exploration Policies for Navigation. ICLR 2019.
* More common tasks for Sec 6.1. Since guided exploration with GPS or satellite images is chosen, there are not many baselines to compare against here. For example, image goal navigation can be used and demonstrated for unseen environments. Thus, the method can be compared with a variety of image goal navigation approaches (e.g. neural topological SLAM). Or, point goal navigation can be compared against DD-PPO (Wijmans et al. ICLR 2019).
    * While it is great to see generalization to unseen environments, it's also important to show that there is no degradation in performance in seen environments (compared to baselines).
* Table 1 only shows a single metric for coverage. The single metric is insufficient, since it only measures distance traveled before collision rather than measuring coverage itself (e.g. the robot could be traversing in circles.
* Please clarify Sec. 6.4 / Appendix E3.2. Overall, it is unclear. For example, is the finetuning done with behavior cloning? How is the graph constructed? Is a graph pre-constructed ie. prior to inference/evaluation time? How dense are the nodes? Do all baselines use the graph for navigation as well?
* The heuristic seems hand-engineered, arbitrary, and brittle. How much tuning is involved with creating the heuristic? How sensitive is the model to the heuristic? e.g. how does performance differ if the weightings of the individual values change in the equation at line 499 appendix

**Quality Of The Limitations Section:**

Limitations are addressed clearly

**Questions For Rebuttal:**

See above.

**Robotics Focus:**

Sufficient demonstration on hardware

**Summary Of Paper:**

The authors propose a method for training a foundation model for robot navigation. They create a large dataset by combining several open research datasets on robot navigation and train a transformer model to perform image goal navigation on the big dataset. To apply this to other downstream applications, they combine the trained model with a diffusion model for sampling subgoals (images) and a heuristic for determining which subgoals to use. The proposed approach is demonstrated to generalize to new environments, new robots, and even new tasks (with finetuning), outperforming all baselines.

**Summary Of Recommendation:**

Overall, I like the premise of the paper a lot. I also think the approach overall seems reasonable (at least Sec. 3). Moreover, the experimental questions that are asked are particularly important and exciting for the community if we can successfully do all those things with a single model. What's holding this paper back for me is the paper clarity (e.g. many important details are left in the appendix, and the appendix seems to be missing details too), and the experiments. Rather than answering many questions fairly quickly, I'd rather see fewer questions answered but with high quality experiments to support the claims. This can be done by demonstrating the claim against stronger baselines and on more tasks (e.g. the guided exploration task seems cherrypicked where there are not many baselines).

---

> ### Author Response · Authors · 2023-08-12
> **Request to Engage**
>
> We hope the new experiment and clarifications helped address your concerns. It would be very useful to us if you could provide your updated feedback, or any additional concerns you may have, to help us improve the paper. The CoRL discussion phase ends **in 3 days**.

---

### Official Review · Reviewer_EDM6 · 2023-07-20

**Confidence:** 4
**Originality:** Good
**Technical Quality:** Good
**Clarity Of Presentation:** Good
**Impact:** 4

**Recommendation:**

Weak Accept: I recommend accepting the paper, but will not argue for my recommendation if the majority of other reviewers have a different opinion.

**Review:**

Strengths:

- The introduction of the paper is effectively crafted, providing a clear and compelling rationale for developing a system capable of generalization in the robotics context. The idea's intuitive appeal is well-explained, giving readers a strong motivation to delve into the proposed approach.
- One particularly intriguing aspect of the research is its focus on the system's adaptability to novel modalities. This unique application warrants further investigation, as it has the potential to yield valuable insights and advancements in the field.
- The website's inclusion of both code and pre-trained weights for the proposed model is undoubtedly beneficial for the wider community. By facilitating straightforward testing and adoption of the architecture, researchers can easily integrate and build upon the work.
- The real-world experiments in the paper serve as a powerful demonstration of the system's utility. Showcasing the efficacy of the proposed approach in practical scenarios is highly commendable and adds credibility to the research findings.

Weaknesses:

- Related work sections mostly outline robotics based foundations models but miss mentioning various relevant papers that perform general visual navigation using GPS goals or goal images.
- Various works that utilize advances in generative models (like diffusion models) to obtain sub goals could be included in the related works section
- Question: the approach mentions that a unique output from ViNT is obtained in the form of a temporal distance. It would be interesting to see how the model could adapt/perform in cases where the goal is not reachable. This could be tested in simple block world examples for simplicity as well. In such cases, the current setup might lead to an ill posed problem.
- The model also includes an output where a sequence of future actions is provided. It would be interesting to see how the model adapts in case of changes in the environment. For example, if certain parameters of the environment are varied the model should be equipped with the ability to modify the previous served action sequence.
- An ablation study of the various components of the proposed approach in the main paper would be helpful for the readers.
- The current model size is relatively large in comparison to the hardware resources available to most of the community. This might make it restrictive in terms of who can use this for actual deployment. Also, the impact of increasing model size is not studied. The tradeoff between the number of parameters and performance could be an interesting addition for the readers. Another smaller model (with slightly inferior performance) could also be provided alongside for those with limited hardware resources.
- The paper specified that the robot is deployed at 4 Hz frequency which might act as a bottleneck in some cases.


**Quality Of The Limitations Section:**

Additional details required

**Questions For Rebuttal:**

Please check the "Strengths and Weaknesses" section for all the questions that need to be addressed.

**Robotics Focus:**

Sufficient demonstration on hardware

**Summary Of Paper:**

The paper proposes ViNT – a foundation model designed for vision-based robotic navigation. It is trained on large and diverse datasets, enabling efficient adaptation to various navigational tasks and outperforming specialist models. The proposed methodology exhibits positive transfer, can explore novel environments, and solve kilometer-scale navigation problems with the ability to accommodate different problem domains. Results also show zero-shot transfer capabilities across new robots.


**Summary Of Recommendation:**

The paper proposed an intuitive idea and is well-presented. The experiments are detailed and exhaustive which makes a strong contribution to the paper. Some minor issues/questions shall be addressed, as highlighted above. I recommend accepting the paper.

---

> ### Author Response · Authors · 2023-08-12
> **Request to Engage**
>
> We hope the new experiment and clarifications helped address your concerns. It would be very useful to us if you could provide your updated feedback, or any additional concerns you may have, to help us improve the paper. The CoRL discussion phase ends **in 3 days**.

---

> ### Comment · Reviewer_ikk2 · 2023-08-15
> **Response to the authors**
>
> I appreciate the authors spending time adding new experiments and giving detailed explanations. I think the authors answered my doubts. I'm happy to raise my rating.

---

### Official Review · Reviewer_ikk2 · 2023-07-24

**Confidence:** 4
**Originality:** Good
**Technical Quality:** Good
**Clarity Of Presentation:** Good
**Impact:** 4

**Recommendation:**

Weak Accept: I recommend accepting the paper, but will not argue for my recommendation if the majority of other reviewers have a different opinion.

**Review:**

The paper describes an interesting problem of training a foundation model for visual navigation. It is currently a scorching topic to train "foundation" models for everything. Naturally, this paper caught wide attention as soon as it appeared on Arxiv. But looking deeply into the paper, it looks more like a stitch work of the authors' two prior works of ViKing and GNM, rather than one groundbreaking "foundation model".

Strengths:
This is an interesting problem and the authors show interesting hardware results.

Weaknesses:
The contribution of this paper is not highlighted. As far as I understand, the main improvement of this paper over the prior work is 1) a diffusion model that generates the sub-goal images, 2) adding other prompt formats. The authors have not shown very convincing results on how these two improvements contribute to the navigation foundation model. Actually, these two improvements are not about the foundation model, but more about how to use the foundation model in different navigation tasks.

**Quality Of The Limitations Section:**

Limitations are addressed clearly

**Questions For Rebuttal:**

1. This paper uses "hundreds of hours" of data to train the model. I have always been curious about how much data it requires to train a robot foundation model. I would love to hear the authors' opinions, do you think the size and diversity of the navigation data is still a bottleneck in this study?
2. Similar to GNM, this work doesn't take into account the difference in camera intrinsics (FoV) and extrinsics (the install angle, etc.). Do you think this affects the generalization ability of the model?
3. When fine-tuning on other types of prompts, i.e. the GPS waypoints and turn-by-turn instructions, the prompts are mapped to a shared space with the vision inputs. I wonder how the authors make sure the token space is shared? After fine-tuning, does the model still work for the visual token?
4. The diffusion sub-goal generation model is a key difference in this paper compared to prior work. But there are not a lot of details about how that is trained. The author mentioned in 6.5 that the model generates invalid sub-goals. I wonder at about how much percent the model generates the reasonable sub-goals? In the video and paper, I could find any visualization about it.
5. One difference of this paper with the GNM is that using a transformer model. How much difference has it made?

**Robotics Focus:**

Sufficient demonstration on hardware

**Summary Of Paper:**

The paper proposes a foundation model for visual navigation tasks. The model is trained with a visual goal navigation task on multiple datasets. The paper also presents a diffusion-based goal generation module, a search-based planning module to enable long-range exploration. The paper also shows the model can take different prompts such as waypoints and turn-by-turn directions.

**Summary Of Recommendation:**

I think the work is interesting and could inspire more work in this direction. I hope the authors be a little more elaborated about its limitations and lessons learned from the process.

---

> ### Author Response · Authors · 2023-08-12
> **Request to Engage**
>
> We hope the new experiment and clarifications helped address your concerns. It would be very useful to us if you could provide your updated feedback, or any additional concerns you may have, to help us improve the paper. The CoRL discussion phase ends **in 3 days**.

---

### Author Response · Authors · 2023-08-15
**Summary of Discussion Phase**

All reviewers acknowledged the premise and value of the contributions of our paper to the field (LWck, Q7NJ), and appreciated the technical quality and clarity of presentation (Q7NJ, EDM6), and thoroughness of empirical analysis and real-world experiments (EDM6, LWck, Q7NJ, ikk2).

The primary concerns raised by the original round of reviews can be grouped as (1) additional analysis and ablations, (2) novelty/contributions, (3) weak baselines, and some additional clarifications and implementation details. We conducted new experiments, report additional metrics, and answers to questions raised by all reviewers as replies to the original reviews.

- [Reviewer EDM6 concerns were addressed with the new hardware results and clarifications, and they agreed to **upgrade to Strong Accept**](https://openreview.net/forum?id=-K7-1WvKO3F&noteId=YJUmuDyJAt9).
- [Reviewer LWck’s concerns about weak baselines/comparison to prior work and improper metrics were addressed with the new results and discussion](https://openreview.net/forum?id=-K7-1WvKO3F&noteId=vsv7svSSTK). They agree that the reported baselines and results are thorough, and we will include these in the revised version. The remaining concern is around clarity of presentation, and we commit to incorporating their suggestions in revising the paper upon acceptance (specifically regarding details about fine-tuning, graph construction etc.) We also highlight that other reviewers found the paper “well-written” and "enjoyable to read" (Q7NJ), and “effectively crafted and compelling” (EDM6).
- Reviewer ikk2 did not respond to our rebuttal for over 12 days. We believe that we have adequately addressed their concerns regarding novelty (discussion below), comparison to GNM (new experiments), and clarification questions.


Summary of changes:

1. **New experiments**:
Reviewers EDM6 and ikk2 requested additional analysis about the model’s performance with unreachable subgoals and improvements over the GNM baseline. We conducted _new real-world experiments_ to analyze this: discussed [here](https://openreview.net/forum?id=-K7-1WvKO3F&noteId=TEEiDVhaAj) and [here](https://openreview.net/forum?id=-K7-1WvKO3F&noteId=u-rGzCYO-T).

2. **Novelty/contributions**: [Quoted from response to ikk2.](https://openreview.net/forum?id=-K7-1WvKO3F&noteId=u-rGzCYO-T)

 3. **Weak baselines**: [Quoted from response to LWck; reviewer responded that it addresses their concern](https://openreview.net/forum?id=-K7-1WvKO3F&noteId=hAMkgPpc1Gh)

---

### Decision · Program_Chairs · 2023-08-30

**Decision:**

Accept (Oral)

**Comment:**

This paper proposes how to learn generalist policy for image goal navigation that generalised (i) across robots using a robot-agnostic actions representation and (ii) new task specifications such as following waypoints or high-level routing commands.

Authors are requested to take the detailed reviewer feedback into consideration. Specifically address the following: (i) improve clarity on the technical gap the paper is trying to fill (ii) why the contribution extends beyond putting two prior works together and (iii) improving the technical presentation of the high-level navigation system.